# Transcranial Magnetic Stimulation Following a Paired Associative Stimulation Protocol Based on a Video Game Neuromodulates Cortical Excitability and Motor Behavior

**DOI:** 10.3390/biomedicines10102632

**Published:** 2022-10-19

**Authors:** Arantzazu San Agustín, Guillermo Asín-Prieto, Juan C. Moreno, Antonio Oliviero, José L. Pons

**Affiliations:** 1Neural Rehabilitation Group (NRG), Cajal Institute, Consejo Superior de Investigaciones Científicas (CSIC), 28002 Madrid, Spain; 2PhD Program in Neuroscience, Cajal Institute, Autonoma de Madrid University, 28029 Madrid, Spain; 3Legs & Walking AbilityLab, Shirley Ryan AbilityLab, Chicago, IL 60611, USA; 4Biomedical Engineering Department, McCormick School of Engineering and Applied Science, Northwestern University, Evanston, IL 60208, USA; 5Mechanical Engineering Department, McCormick School of Engineering and Applied Science, Northwestern University, Evanston, IL 60208, USA; 6Gogoa Mobility Robots S.L., 48220 Abadiño, Spain; 7FENNSI Group, Hospital Nacional de Parapléjicos, SESCAM, 45004 Toledo, Spain; 8Advanced Neurorehabilitation Unit, Hospital Los Madroños, 28690 Brunete, Spain

**Keywords:** Transcranial Magnetic Stimulation, paired associative stimulation, video game, reaction time, corticospinal plasticity, movement-related cortical stimulation, fatigue

## Abstract

Transcranial Magnetic Stimulation (TMS) can be used to modulate cortico-spinal excitability following a paired associative stimulation (PAS) protocol. Movement-related cortical stimulation (MRCS) is a PAS protocol based on the synchronization of a single-pulse TMS with a movement task. However, plasticity and motor performance potentiation due to MRCS has been related exclusively to single-movement tasks. In order to unveil the effects of an MRCS protocol in complex movements, we applied PAS synchronized with a movement-related dynamic task (MRDT) with a customized video game. In 22 healthy subjects, we measured the reaction time (RT), trajectory error (TE), and the number of collected and avoided items when playing the custom video game to evaluate the task motor performance. Moreover, we assessed the recruitment curve of Motor Evoked Potentials (MEPs) with five different intensities to evaluate the motor corticospinal excitability. MEPs were recorded in *Abductor Pollicis Brevis* (APB) and *Abductor Digiti Minimi* (ADM), before, right after, and 30 min after the PAS intervention, in an active versus sham experimental design. The MRCS PAS intervention resulted in RT reduction, and motor corticospinal excitability was modulated, reflected as significant MEP amplitude change at 110% RMT intensity in ADM and at 130% RMT intensity in APB. RTs and ADM MEP amplitudes correlated positively in specific time and intensity assessments. We conclude that the proposed PAS protocol facilitated RT performance in a complex task. This phenomenon might be useful to develop neurorehabilitation strategies with complex movements, similar to activities of daily living.

## 1. Introduction

Synaptic plasticity is the basis of learning and memory processes at the cellular level [1]. Spike-timing-dependent plasticity (STDP) has been experimentally observed by the synchronized activation of two synaptically connected neurons, modulating the synaptic strength [2]. These plastic mechanisms depend on the relative timing and temporal order of pre- and post-synaptic depolarization, and they could lead to Long Term Potentiation (LTP) or Long Term Depression (LTD) [3,4,5,6,7,8,9].

The emergence of non-invasive brain stimulation (NIBS) techniques such as Transcranial Magnetic Stimulation (TMS) enabled the study of neuromodulation in the human brain [10,11]. LTP associative models studied in vitro have been proven in vivo by using paired associative stimulation (PAS) protocols in humans, i.e., pairing two stimuli synchronized in time and neuronal pathways. The first PAS protocol was based on the synchronization between the low-frequency peripheral stimulation of somatosensory afferent pathway 25 ms before single-pulse TMS (spTMS) over the contralateral primary motor cortex (M1) [12]. This paradigm assumes a synchronization between sensory afferences and the TMS pulse over M1, resulting in an enhancement of the cortico-spinal pathway excitability and plastic changes [13]. These modulatory effects have been shown to correlate with improved motor performance in healthy volunteers [14] and in patients with neurological disorders, such as schizophrenia [15], chronic traumatic tetraplegia [16], and stroke [17,18].

Furthermore, the PAS paradigm has been applied associating spTMS to the endogenous activation of M1, triggered by thumb abduction during a simple reaction time (RT) task [19]. This movement-related cortical stimulation (MRCS) protocol resulted in an enhancement in corticospinal excitability and improved RT [20]. MRCS provides a method that induces plastic changes without peripheral electrical stimulation and opens the possibility to pair TMS with any cortical activity triggered by a task, in order to enhance the task performance [21].

MRCS PAS has been exclusively applied to simple movement tasks. Our goal is to investigate if PAS also leads to facilitating performance in more complex motor tasks, which can be considered closer analogs to activities of daily living (ADLs). To this end, we report on the design of a new MRCS PAS protocol, framed in a video game scenario, and aimed at performing a functional motion goal. This allowed us to assess the reaction, precision, and directionality performance in dynamic motor behavior. We studied the PAS effects on motor performance and corticospinal excitability before, immediately after, and 30 min after the PAS intervention and analyzed the relationship between both outcome measures.

Our ultimate goal is to assess the practical application of MRCS PAS to the neuromodulation of complex movements involved in ADLs. Therefore, this study can be regarded as an intermediate step towards this goal. Our results showed the potential for improving functional and dynamic movements, and thus pave the road for future studies towards the application of these protocols to motor rehabilitation.

## 2. Materials and Methods

### 2.1. Subjects

Twenty-two healthy volunteers (ages 21–63; 9 females, 13 males) participated in this study. Eleven subjects participated in the experimental group where active stimulation was delivered (Active group) and eleven subjects participated in the control group where sham stimulation was applied (Sham group). In order to avoid learning and carry-over effects, subjects were assigned exclusively to one of the groups. None of the subjects had past or current neurological disorders and they did not have any pharmacological treatment on the day of the experiment. All participants gave informed consent before the experimental intervention. The study was conducted in accordance with the Declaration of Helsinki and The Spanish National Research Council (CSIC) Ethics Committee approved all procedures.

### 2.2. Transcranial Magnetic Stimulation

TMS was applied using a figure-of-eight double 70 mm remote control coil and a single-pulse, monophasic stimulator (Magstim 200^2^ stimulator). TMS was applied to induce a current in the posterior-to-anterior direction. 

During the corticospinal excitability assessment, TMS was used to elicit motor evoked potentials (MEPs) in the *Abductor Pollicis Brevis* (APB) and *Abductor Digiti Minimi* (ADM) muscles. The subjects were seated in a comfortable armchair and their dominant hand was placed on the armrest. First, we conducted motor mapping to identify the cortical location where thumb stimulation (APB muscle) was optimal (hot spot). We started with a TMS pulse of 50% of the maximum stimulator output intensity, and a localization of 10 mm anterior and 20 mm right or left to bregma, depending on the subject’s dominant hand. This point was marked on the subject’s scalp to ensure the same coil position throughout the experiment. Subsequently, the Resting Motor Threshold (RMT) was estimated as the stimulation intensity that evoked, 5 out of 10 times, an MEP with a peak-to-peak amplitude of at least 50 µV while the muscle was at a resting state [22].

Then, 50 MEPs were elicited at different intensities (10 pulses of 90%, 100%, 110%, 120%, and 130% of the RMT), in a randomized order (input–output recruitment curve). We also obtained a recruitment curve for the ADM muscle (adjacent muscle not primarily involved in the motor task) by using the same hot spot and intensity as for APB. Thus, the ADM MEPs were obtained in a suboptimal condition. We measured the average peak-to-peak amplitude of the MEP at each intensity.

During the PAS intervention, for the Active group, TMS was used to induce plastic changes synchronized with the M1 activation triggered by complex motor task performance. TMS was delivered at 110% of the RMT intensity over the APB hot spot while participants were performing the complex motor task. For the Sham group, a figure-of-eight double 70 mm TMS coil which did not deliver active stimulation was placed on the subjects’ APB hot spot location, and another figure-of-eight double 70 mm TMS coil that delivered pulses was held perpendicularly to the scalp so that no current was induced in the brain. The perpendicular TMS trigger was synchronized with the complex motor task performance, like the Active group. Subjects were blind to the kind of TMS they received (Active or Sham).

### 2.3. Electromyography (EMG) Recordings

We used bipolar surface EMG electrodes to record APB and ADM activity. The EMG signals were amplified by g.USBamp biosignal amplifier (g.tec), 50 Hz noise was removed with a notch filter, digitized at a rate of 2400 Hz, and stored for offline analysis. During MEP assessment, we monitored the bioelectrical state of both muscles, with the aim of delivering an spTMS in the muscle at a resting state. During signal analysis, we discarded the MEPs that previous to the TMS artifact showed myographic activation.

### 2.4. Movement-Related Dynamic Task (MRDT)

The video game prompted the subjects to track trajectories depicted via collectible items (“bottles”), while avoiding other objects (“tubes”) presented on the screen, by controlling the movement of the character (“gyrocopter”). 

The MRDT consisted of five different trajectories, distributed randomly across 30 trials in two intervention blocks. Each trial lasted 10 s and required six repetitive movements. Thus, the subject had to complete 360 short APB contractions over 10 min. The user was able to command the up and down movement of the “gyrocopter” on the screen by moving the instrumented thumb to the left or right, respectively. An absolute position sensor was used to monitor the thumb’s angle, which was acquired by a custom-designed electronic board, sent to a BeagleBone Black (BBB) board (CAN to UDP converter), and then to the main computer, where the visual paradigm was run in MATLAB [23] (Figure 1a).

Each trial consisted of four phases (Figure 1b): (1) Resting phase: the subject had to wait for the “gyrocopter” to be displayed (1500 ms); (2) pre-movement phase: the subject had to keep the “gyrocopter” between the dotted lines (1500 ms); (3) movement phase: the subject moved the “gyrocopter” to collect the target items (“bottles”) on the screen while avoiding the “tubes” (5000 ms); (4) feedback phase: the subject was given feedback, i.e., their performance for the trial (2000 ms). The movement phase’s beginning is visually cued when the “gyrocopter” enters the “sky”, i.e., the screen background changes from white to blue.

Subjects were asked not to move until the movement phase in order to collect bottles and avoid tubes. The aim of the exercise was to train the RT and the maximum collection of items following the best linear path between bottles.

The outcome measures for each trial were: (1) RTs: the elapsed time from the visual cue and the first EMG activation of APB, assessing the time-accuracy motor performance; (2) trajectory error (TE): the error between the angular position of the subject and the ideal angle, assessing trajectory-accuracy motor performance; and (3) the number of collected and avoided items, assessing the object-directed motor performance.

During the two intervention blocks, 30 trials per block, an spTMS was delivered at the onset of each movement phase (either Active or Sham). As this movement task has a pre-movement phase, the participants were instructed to move at the onset of the movement phase. They predicted the specific time when they should move to correctly perform the task. Thus, we hypothesized that the M1 cortical area would be activated around 25 ms before the movement phase and would remain active as long as the subject continued to move. In order to synchronize the M1 activity with the spTMS, we chose the predicted movement time (onset of movement phase) when M1 would be activating or remaining active.

### 2.5. Experimental Design

This is an active/sham parallel experimental design (Figure 2). The experimental session began with an assessment of basal skills in task performance during 30 trials and the cortico-spinal excitability assessment of APB and ADM of the dominant hand by an MEP recruitment curve. MEP assessment was repeated twice in order to ensure stability and reproducibility (Pre-Pre-Assessment and Pre-Assessment). Afterwards, the PAS intervention was conducted in two blocks of 30 trials each (the first intervention and the second intervention block; Int1 and Int2, respectively). TMS was applied at the onset of the movement phase. The motor task performance and MEP recruitment curve were assessed again, immediately and 30 min after the intervention (Post-Assessment and Post-30’-Assessment). 

### 2.6. Statistical Analysis

RT, TE, and the number of collected and correctly avoided items were used to rate MRDT motor performance improvement. We transformed the data into absolute values for the RT parameter and normalized TE, the number of collected items, and the number of avoided items by the formula (individual value—mean/standard deviation). As the task was performed during assessment and intervention, we compared the motor indices between each experimental time (Pre-Assessment, Int1, Int2, Post-Assessment, and Post-30’-Assessment). A repeated-measures ANOVA with the factors Time (Pre-Assessment, Int1, Int2, Post-Assessment, and Post-30’-Assessment) × Group (Active group and Sham group), and post hoc analysis with Bonferroni correction was calculated. Greenhouse–Geisser correction was used when the sphericity assumption was not met. 

The corticospinal pathway excitability was assessed by MEP amplitude for all intensities at each assessment time. We normalized the raw MEP amplitude (µV) (individual value—mean/standard deviation) and calculated a repeated-measures ANOVA with Time (Pre-Pre-Assessment, Pre-Assessment, Post-Assessment, and Post-30’-Assessment), Intensity (90%, 100%, 110%, 120%, 130% RMT), and Muscle (APB and ADM) as intra-subject factors and Group (Active group and Sham group) as an inter-subject factor. Post hoc analysis using Bonferroni correction was applied. Again, Greenhouse–Geisser correction was applied when the sphericity assumption was not met.

The correlation between task performance and corticospinal excitability was analyzed by the Pearson correlation coefficient. We correlated task performance indices that were statistically different between groups, with TMS intensities statistically different between Times. Finally, we calculated the correlation of RT in the first intervention block with the MEPs in the Post-Assessment and Post-30’-Assessment times at 110% and 130% RMT intensities. All data were transformed into proportional data based on Pre-Assessment time. Grubbs’ test was applied to detect and delete outlier scores. 

The statistical procedures were calculated with the aid of statistical software IBM SPSS for every statistical analysis [24] and significance was set to *p* < 0.05.

## 3. Results

There are no significant differences between the ages of the subjects in the Active group (Mean = 28.54 yrs., standard deviation (SD) = 6.28) and the Sham group (Mean = 29.72 yrs., SD = 11.92).

### 3.1. Behavioral Results

Comparing performance indices between times and groups, we calculated the PAS intervention effects on the task performance ability. Mean RT at baseline was similar between the Active and Sham groups (Active: Mean = 0.262 s, standard error (SE) = 0.072 s; Sham: Mean = 0.282 s, SE = 0.069 s; *p* > 0.05). In RT data, a significant main effect of Time F(4, 80) = 9.134 (*p* = 0.000), Group F(1, 20) = 5.984 (*p* = 0.024), and an interaction effect of Time x Group F(4, 80) = 2.637 (*p* = 0.040) was found.

The post hoc analysis (Bonferroni correction) revealed that the Active group showed a significantly reduced RT with respect to the Pre-Assessment at all assessment times: Int1 (Mean = 0.128 s, SE = 0.019 s, *p* = 0.000), Int2 (Mean = 0.159 s, SE = 0.026 s, *p* = 0.001), Post-Assessment (Mean = 0.194 s, SE = 0.022 s, *p* = 0.008), and Post-30’-Assessment (Mean = 0.169 s, SE = 0.025 s, *p* = 0.001). Compared to the first intervention time, the RT at Post-Assessment time was significantly reduced (*p* = 0.006). For the Sham group, only the Post-30’-Assessment time (Mean = 0.172 s, SE = 0.022 s, *p* = 0.009) showed a reduced RT with respect to the Pre-Assessment. The Active group’s RT was reduced significantly compared to the Sham group (Sham: Mean = 0.245 s, SE = 0.019 s, *p* = 0.000) in the first intervention block (D = 0.117 s, SE = 0.026 s) (Figure 3).

With respect to the TE, both the Active and Sham groups significantly improved trajectory path adjustment over time. A significant main effect in Time F(2593, 41,484) = 122.961 (*p* = 0.000) was found, while no significant differences were found between groups. For the number of collected items, both groups significantly changed over time; i.e., a significant main effect of Time F(2739, 46,505) = 95.337 (*p* = 0.000) was found. However, no changes between groups were observed. Likewise, with respect to the number of avoided items, a significant main effect of Time F(2809, 53,363) = 31.854, *p* = 0.000 was found, although no significant differences were shown between groups.

In summary, the obtained results demonstrated that starting at the first intervention block and continuing through each subsequent assessment point, the Active group was able to perform shorter RTs. The time-accuracy motor performance defined by RTs was improved by PAS intervention, whereas trajectory-accuracy performance, defined by TEs, and object-directed performance, defined by the number of collected and avoided items, were not affected by PAS intervention.

### 3.2. Cortico-Spinal Excitability

Figure 4 shows MEP amplitudes (mean ± SE) for both muscles in the two groups. First, MEP amplitudes were normalized by mean and SE (x’ = (x − μ)/σ) and analyzed by a mixed ANOVA of four factors (i.e., Muscle, Time, and Intensity as intra-subject factors and Group as an inter-subject factor). Significant main effects of Muscle (F(1, 20) = 7.855, *p* = 0.011), Time (F(3, 60) = 3.565, *p* = 0.019), and Intensity (F(2988, 59,768) = 298.606, *p* = 0.000) were found, as well as Muscle x Intensity (F(1553, 31,051) = 6.456, *p* = 0.008), Time x Intensity (F(4979, 99,578) = 2.483, *p* = 0.037), and Muscle x Time x Intensity x Group (F(5510, 110,193) = 3.030, *p* = 0.011).

The post hoc pair comparison (Bonferroni correction) revealed that in the Active group at Pre-Assessment time, the MEP amplitude induced in the APB muscle at 130% RMT (Mean = 2.456, SD = 0.388) was significantly higher than at the Post-Assessment time (Mean = 0.977, SD = 0.263, *p* = 0.021) (Figure 4a).

In contrast, at the Post-Assessment time, the MEP amplitude induced in ADM muscle at 110% RMT (Mean = −0.451, SD = 0.102) was significantly lower than in Post-30′-Assessment (Mean = 0.056, SD = 0.197, *p* = 0.047) (Figure 4b).

There are no significant differences between time points in the Sham group (Figure 4c,d).

These results might indicate that the TMS intervention synchronized with the MRDT neuromodulated the motor excitability, and this is present in the APB muscle after the task assessment at 130% RMT intensity and in the ADM muscle after 30 min assessment, at 110% RMT intensity. Therefore, the presence of PAS effects over corticospinal excitability is very specific regarding the TMS intensity in the MEP assessment.

### 3.3. Correlation between RTs and MEPs

The application of MRCS PAS induced changes in motor performance, and in the excitability of cortico-spinal pathways found at certain intensities. TMS neuromodulation can be attributed to the correlation between significant changes in behavior and corticospinal excitability.

For the Active group, we found positive linear correlations (Figure 5) between the RT of the Int1 time (with significantly different RT conditions between groups) and the MEP amplitude induced in ADM muscle in Post-Assessment at 110% RMT intensity (r = 0.775, *p* = 0.008) and 130% RMT intensity (r = 0.668, *p* = 0.035), as well as at Post-30’-Assessment time at 130% of RMT intensity (r = 0.634, *p* = 0.049). This indicated that the lower the MEP amplitude in ADM muscle, the shorter the RT.

For the Sham group (Figure 6), a tendency towards negative linear correlation was observed between the RT of the Int1 and the amplitude of MEP induced in the ADM muscle at Post-Assessment at 110% RMT intensity (r = −0.495, *p* = 0.122), at 130% of RMT intensity (r = −0.575, *p* = 0.064), and at Post-30′-Assessment time at 130% of RMT (r = −0.559 *p* = 0.074). This indicated that the higher the MEP amplitude in the ADM muscle, the shorter the RT.

No correlation was found between the RT of the Int1 and the APB MEPs. No other correlations between RT and MEP amplitude were found.

## 4. Discussion

The main goal of this study was to evaluate the effects of an MRCS PAS intervention in a complex movement-related exercise. The main result found was an improvement in task performance (i.e., RT reduction) in the Active group, in contrast to the Sham group. Other outcome measures, i.e., the trajectory-accuracy and object-directed performance, improved over time, suggesting a learning effect; however, the Active group did not show improved performance compared with the Sham group. Moreover, PAS-related neuromodulation was found at certain intensities in motor cortex excitability assessments and correlated with RT results.

### 4.1. PAS Effects on Behavioral Indices

#### 4.1.1. RT Index

In the Active group, the PAS intervention reduced the RT compared to the baseline in both intervention blocks, suggesting that the intervention induces an immediate shift in behavior. Furthermore, a significant reduction in RT was observed immediately after the intervention, indicating short-term effects of the intervention on motor performance.

The reduction in RT in both intervention blocks does not necessarily imply plasticity. The RT was around 120 ms shorter in the Active group than in the Sham group in the first intervention. However, the difference in RT in the second intervention was reduced to 70 ms. A possible explanation is that the TMS pulse facilitated the motor execution of the task without inducing neuronal connectivity changes. However, the reduction in RT after the intervention likely implies induction of corticospinal plasticity, which is built up over time.

#### 4.1.2. TE and the Number of Collected and Avoided Items

The trajectory-accuracy and object-directed motor performance, measured respectively by the TE and the number of collected and avoided items, were not affected significantly by the PAS intervention. The neuromodulation by MRCS PAS was induced at the M1 cortex, and its effects have been assessed by the RT outcome measure. The M1 brain area is the final stage in processing voluntary movements and its function is involved in the initiation and inhibition of these movements [25]. The RT is precise in evaluating M1 function because the participants have to inhibit the movement until a cue triggers it. M1 function is not specifically the control of fine movements nor the planning of movement. For this reason, we believe that more research is needed on MRCS PAS to induce plastic changes in areas responsible for precision and directionality movements, such as the ventral premotor cortex (PMv). PMv is related to goal-directed manual actions and the control of fine finger movements [26,27]. This might lead to a PAS methodology that enhances trajectory accuracy and object-directed complex movements for the facilitation of ADLs.

### 4.2. PAS Effects on MEPs

#### 4.2.1. APB Muscle

The changes in the excitability of the motor corticospinal pathway, measured by the amplitudes of the MEPs, showed that PAS induced a depression at Post-Assessment when 130% of the RMT intensity was applied to the APB (the target of the stimulation and main muscle performing the motor task). At least two different factors may contribute to the excitability changes, namely plastic changes and fatigue mechanisms.

We suggest that fatigue was likely occluding MEP facilitation in the motor corticospinal excitability assessment. Fatigue in the motor corticospinal pathway correlates with decreased MEP amplitude and the possible mechanisms for this change are the reduction in synaptic transmission at cortical and cervicomedullar levels [28,29,30,31,32,33]. MRDT required 360 short APB contractions that could be responsible for generating motor fatigue. In our experiment, the APB muscle had higher active participation during the task than ADM, so it is conceivable that the APB and its corticospinal connections were more fatigued than the ADM. When a PAS protocol and a fatigue task, or vice versa, are consecutively conducted, processes of potentiation due to the intervention and depression due to fatigue are occluded by the opposing effects, rendering no significant change in MEP between baseline and posterior assessments [34]. This agrees with our results, which suggest that PAS plasticity induction was not resulting in MEP enhancement at the first four TMS intensities. At the highest intensity (130% of RMT), we observed a reduction in the MEP amplitude after the intervention. This might suggest that fatigue was overcoming the effects of PAS-induced plasticity at this stimulation intensity.

We hypothesize that PAS plasticity was important to induce task performance improvement, but the excitability improvement was not apparent in the MEP due to the contribution of fatigue. MRCS PAS might be inducing, at the same time, corticospinal excitability potentiation, fatigue mechanisms, and RT improvements. Nevertheless, more research in fatigue along with PAS interventions is needed to help throw light on this preliminary interpretation.

#### 4.2.2. ADM Muscle

MRCS PAS-induced motor excitability changes on ADM were only evident at around-threshold intensity (110% of RMT). This MEP difference correlated positively with RTs exclusively in the Active group. This suggests that the PAS was the main mechanism to produce both phenomena: RT reduction and ADM excitability modulation at 110% RMT. However, reduced MEPs were correlated with shorter RTs, indicating that PAS was likely contributing to behavioral potentiation as much as MEP modulation. 

The differential effects on ADM and APB cortical excitability were probably due to PAS effects not being muscle-specific.

### 4.3. Why Are the MEP Effects Different Depending on the TMS Pulse Intensity?

Patton and Amassian characterized the electrical stimulation descending volley through the direct recording of the motor cortex pyramidal tract in cats and primates [35]. They suggested two types of waves in the induced response: the initial volley produced by the direct stimulation of the pyramidal neuron axon, called the D wave, and the later volleys produced by the indirect activation of pyramidal corticospinal neurons when interneurons were activated, called I waves. Other studies found that while TMS recruits I waves at threshold intensity, at supra-threshold intensity TMS activates both the cortico-cortical and the pyramidal nervous cells [36,37,38,39,40,41,42].

We found MEP modulation in the ADM muscle at 110% RMT and MEP depression in the APB muscle at 130% RMT. As for changes in the ADM MEP, we suggest that PAS was indirectly activating corticospinal neurons at 110% of RMT. The PAS facilitation was not shown when TMS pulse intensity was higher, probably because higher intensities stimulate a larger number of corticospinal neurons. Thus, this PAS induction could affect cortico-cortical plasticity to a greater extent.

Regarding the depression of APB at 130% RMT, as discussed above, the PAS effects may not be observed due to fatigue. The D wave contributes to the MEP amplitude and is sensitive to fatigue [43], therefore, it is conceivable that fatigue could affect mostly the MEP obtained at higher TMS intensities when the pyramidal neuron axons are recruited. Thus, fatigue may affect the corticospinal cells more than the cortico-cortical neurons.

In conclusion, we suggest a mechanism in which plastic changes are induced by PAS (demonstrated by behavioral effects), and where the fatigue affecting mostly pyramidal neurons may explain why these effects are not present in the MEP amplitude changes. However, controlling the fatigue factor in future research is needed to support this suggestion.

### 4.4. Methodological Considerations and Limitations of the Study

The results reported herein should be considered in light of some limitations that should be addressed in future research.

Fatigue assessment was not included in our research; therefore, we could not precisely determine to what extent this factor is playing a role. Since fatigue is likely a fundamental factor influencing the results, its evaluation is necessary for further studies.

The RT index in the Active group significantly improved compared to the Sham group, whereas other behavioral outcomes remained unaltered. In future studies, the assessment of the role of different cortical areas, such as PMv, is required. This would allow a more informed analysis of relations between behavioral indices and cortical functions and enable the development of function-specific PAS interventions.

Finally, the small sample size must be considered as a limitation of this study when interpreting statistical results. Future investigations will consider the application of this PAS protocol in a larger sample size for more accurate estimations.

## 5. Conclusions

We conclude that the MRCS PAS protocol had an immediate and short-term motor performance effect on time-accuracy movement in a complex task. The plastic changes in motor excitability induced by this protocol were found and correlated to RTs with short- and long-lasting timing. 

Moreover, we demonstrated that the MRCS PAS protocol’s effects on corticospinal excitability are not muscle-specific when paired with a complex task. PAS induced changes in muscles adjacent to the PAS target muscle. We suggested that fatigue might interfere with the PAS protocol, occluding the possible optimal long-lasting effects in corticospinal excitability of the main muscle executing the task and that PAS is likely contributing to increasing that fatigue, despite simultaneously enhancing RTs.

Overall, we conclude that the applications of MRCS PAS can be beneficial for RT facilitation in complex tasks; however, an experimental design that considers fatigue and other specific cortex locations is needed for future validations and function-specific PAS protocols.

## Figures and Tables

**Figure 1 biomedicines-10-02632-f001:**
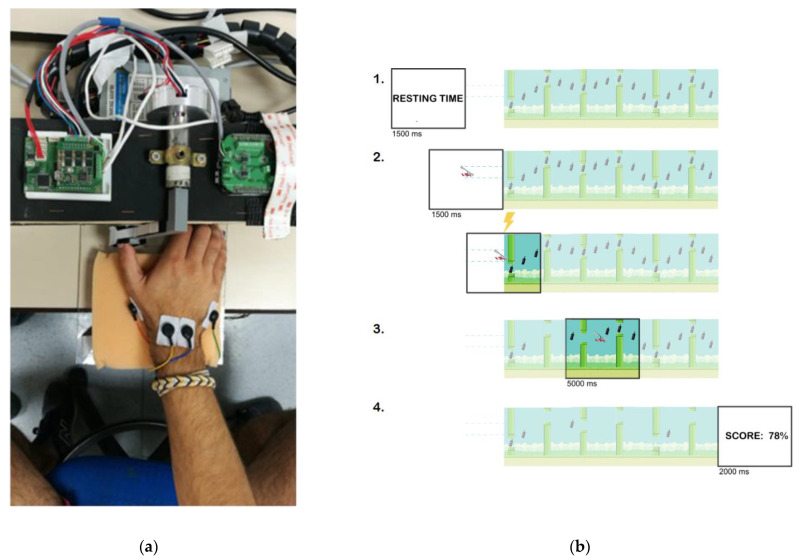
**MRDT performed by a video game.** (**a**) Customized video game device with a position sensor, electronic board, and BBB board. (**b**) Task phases of MRDT: 1. Resting phase; 2. pre-movement phase; 3. movement phase; 4. feedback phase. TMS application time is indicated by the lightning signal.

**Figure 2 biomedicines-10-02632-f002:**
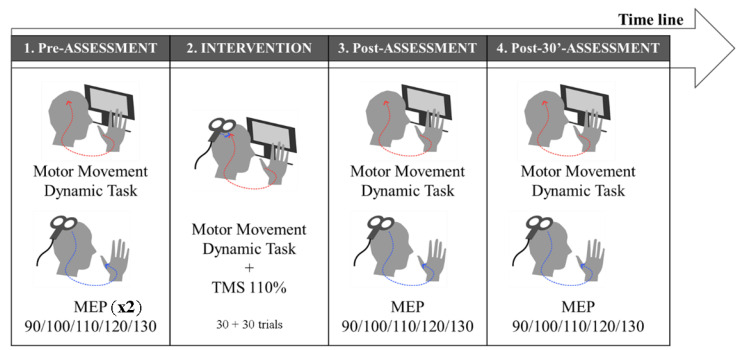
**Experimental timeline design:** 1. Pre-assessment: motor performance indices and the MEP recruitment curve assessment, evaluated twice. 2. Intervention: we applied the PAS protocol synchronizing thumb movement with a TMS pulse. 3. Post-Assessment: we assessed motor performance indices and the motor pathway excitability changes. 4. Post-30’-Assessment: 30 min after the intervention, we assessed the motor performance indices and motor pathway excitability.

**Figure 3 biomedicines-10-02632-f003:**
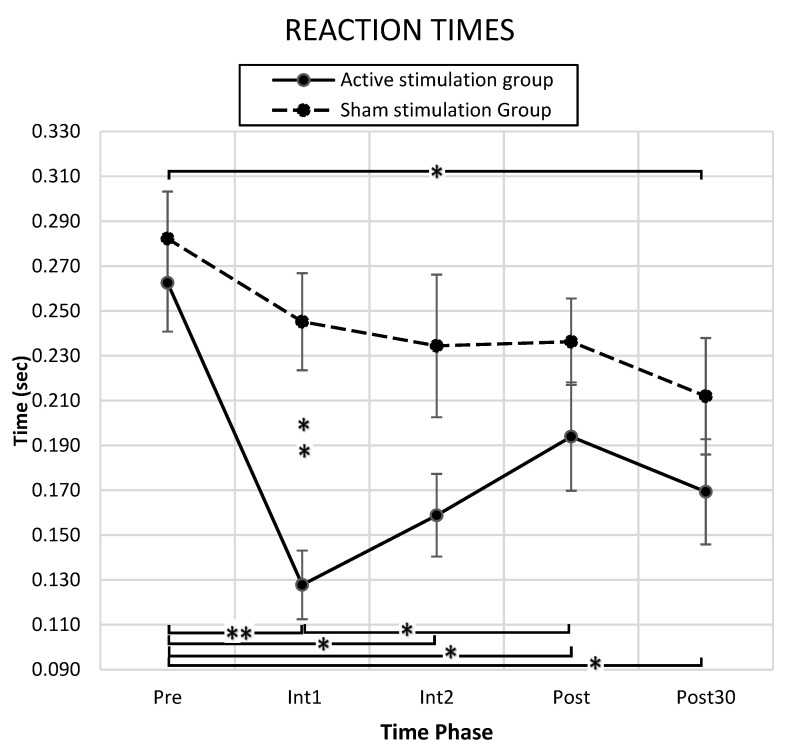
**RT assessments**. Mean and SE of the elapsed time from the cue until the first movement, defined as RT, for Active and Sham groups at each experimental time point: Pre-Assessment (Pre), first intervention block (Int1), second intervention block (Int2), Post-Assessment time (Post), Post-30’-Assessment (Post30). * corresponds to significance between *p* = 0.05 and *p* = 0.01; ** correspond to significance below *p* = 0.01.

**Figure 4 biomedicines-10-02632-f004:**
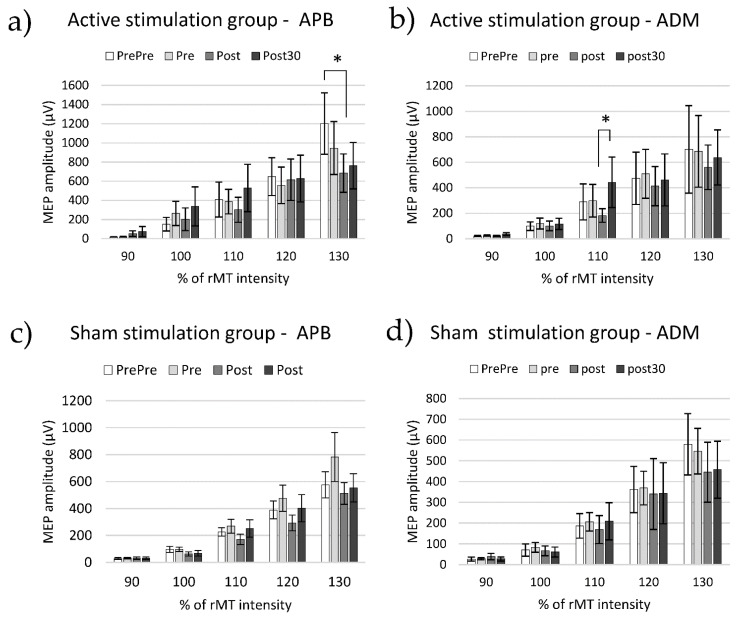
**Recruitment curves**. Raw data of Mean, SE, and significant differences of the peak-to-peak MEP amplitude for APB and ADM muscles in Active and Sham groups, and the different experimental times at each intensity level. Recruitment curves of (**a**) APB muscle in Active group, (**b**) ADM muscle in Active group, (**c**) APB muscle in Sham group, (**d**) ADM muscle in Sham group. * corresponds to significance between *p* = 0.05 and *p* = 0.01.

**Figure 5 biomedicines-10-02632-f005:**
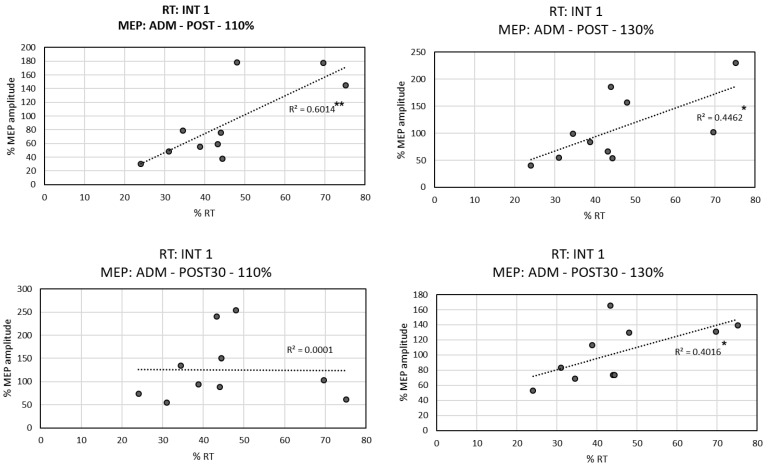
Active group correlation between the % of MEP amplitude and the % of RT based on Pre-Assessment time. Simple linear regression and the coefficient of determination of the correlation of behavior RT performance in the first intervention with MEP amplitudes of ADM in Post- and Post-30’-Assessment at 110% and 130% RMT. R and *p*-value are reported in the results. * corresponds to significance between *p* = 0.05 and *p* = 0.01; ** correspond to significance below *p* = 0.01.

**Figure 6 biomedicines-10-02632-f006:**
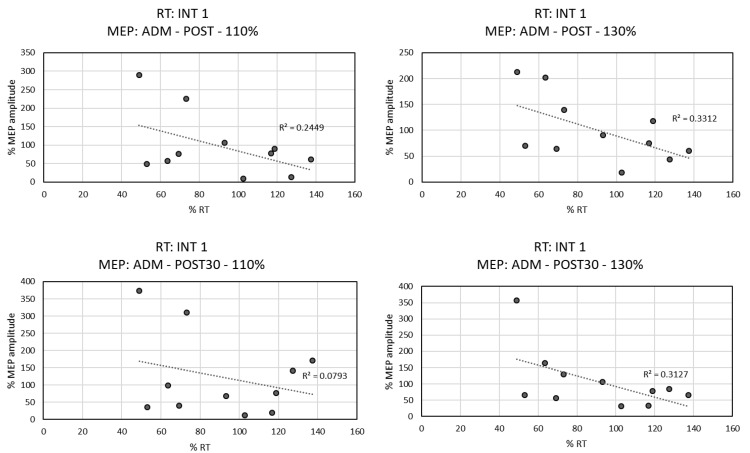
Sham group correlation between % of MEP amplitude and % of RT based on Pre-Assessment time. Simple linear regression and the coefficient of determination of the correlation of behavior RT performance in the first intervention with MEP amplitudes of ADM in Post- and Post-30’-Assessment at 110% and 130% of RMT.

## Data Availability

Data supporting reported results are available on request.

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
