# Peer review of "Transcranial Magnetic Stimulation Following a Paired Associative Stimulation Protocol Based on a Video Game Neuromodulates Cortical Excitability and Motor Behavior"

_biomedicines, 2022, doi:10.3390/biomedicines10102632_

Round 1
Reviewer 1 Report
The manuscript is well written about the effects of the PAS on the processing ability for the movement task.
The minor comment is below.
Minor comment
The authors use a lot the term “fatigue” in the Discussion section. What does the neuronal “fatigue” mean at the molecular biological level? Down regulation of the membrane molecules such as ion channels, internalization, modal-shift of the synaptic activity (excitatory? Inhibitory?), dysfunction of the glial activity, or other else?
Author Response
Response to Reviewer 1 Comments
Point 1: The authors use a lot the term “fatigue” in the Discussion section. What does the neuronal “fatigue” mean at the molecular biological level? Down regulation of the membrane molecules such as ion channels, internalization, modal-shift of the synaptic activity (excitatory? Inhibitory?), dysfunction of the glial activity, or other else?.
Response 1:
The fatigue phenomenon we are refering to in the Discussion section is the one that reduces the MEP amplitude after intense muscle contraction. This type of fatigue, as we mentioned in our article in references 28-34, has been studied by different authors such as Di Lazzaro et al. or Brazil-Neto et al. among others.
The neuronal changes that occur when fatigue develops have been the subject of debate for many years (Di Lazzaro, 2003). In general, this fatigue is related to failure of nerve impulse conduction in the neuromuscular junction. However, as the reviewer rightly pointed out when mentioning synaptic activity, the explanation of these mechanisms has also focused on changes in synaptic transmission at cortical and cervicomedullar levels (Gandevia, 1999). These changes are explained relative to synaptic strength rather than the change from excitatory to inhibitory or vice versa. Specifically, the reduction in MEP amplitudes due to TMS application are compatible with effects related to the accumulation and depletion of neurotransmitters (Brasil-Neto et al., 1993).
Regarding the reviewer’s observation, we included in the Discussion section an especification in defining fatigue at cellular level (line 380-381).
References
Di Lazzaro, V., Oliviero, A., Tonali, P. A., Mazzone, P., Insola, A., Pilato, F., ... & Rothwell, J. C. (2003). Direct demonstration of reduction of the output of the human motor cortex induced by a fatiguing muscle contraction. Experimental brain research, 149(4), 535-538.
Gandevia, S. C., Petersen, N., Butler, J. E., & Taylor, J. L. (1999). Impaired response of human motoneurones to corticospinal stimulation after voluntary exercise. The Journal of physiology, 521(Pt 3), 749.
Brasil-Neto, J. P., Pascual-Leone, A., Valls-Solé, J., Cammarota, A., Cohen, L. G., & Hallett, M. (1993). Postexercise depression of motor evoked potentials: a measure of central nervous system fatigue. Experimental Brain Research, 93(1), 181-184.

Reviewer 2 Report
The mansucript is well-written. Suggested revisions are listed below
-as far as i understand, Authors chose a parametric test for analysis. Did they check for normal distibution? Did they transform their data? They should clarify this decision, as due to limited sample size, a non-parametric test would be possible more reasonble.
-minor spellin errors should be corrected
Author Response
Response to Reviewer 2 Comments
Point 1: as far as i understand, Authors chose a parametric test for analysis. Did they check for normal distibution? Did they transform their data? They should clarify this decision, as due to limited sample size, a non-parametric test would be possible more reasonble.
Response 1:
As a result of the reviewer's comment, we have revised the assumptions in our data necessary to apply the mixed ANOVA in our study to correct the description of statistic procedures in the article. Regarding the outcome-measures, there are 5 different data in this study and we describe here the normal distribution of each parameter:
- Reaction Time (RT): this variable meets the assumption of normality. These data was transformed into absolute values because of the characteristics of the parameter (line 198-199).
- and 4. Trayectory Error (TE), Number of collected items and Number of avoided items: we analysed the normal distribution of these parameters and found that the sample was not following a normal distribution. Thus, we have transformed the sample in the same way as we did for the MEPs, i.e., calculating individual value – mean / standard deviation. Then, we applied the same mixed ANOVA and we found the same significances for the 3 parameters.
We have included in the article the description of the transformation (line 199-201) and the updated ANOVA indices: for the TE parameter (line 246-247), collected items (line 250-251) and avoided items (line 254-255)
- MEP amplitude: The MEP amplitudes (µV) were normalized by calculating the following formula: (individual value – mean / standard deviation) (line 207-208). We would like to take this opportunity to correct as well the F indices, when the sphericity assumption was not met (line 277-280).
In this new version, we have also corrected the ANOVA indices for the 5 parameters, using the Greenhouse-Geisser correction when sphericity is not met. We have included this specification in the Methods section (line 206 and 213-214).
Point 2: minor spellin errors should be corrected
Response 2:
We found and correct 2 spelling errors in line 225 and 226.
